# Identification of dynamical changes of rabies transmission under quarantine: Community-based measures towards rabies elimination

**Kristyna Rysava** *, **Michael J. Tildesley**

The Zeeman Institute for Systems Biology & Infectious Disease Epidemiology Research, School of Life Sciences and Mathematics Institute, University of Warwick, Coventry, United Kingdom

* Kristyna.Rysava@warwick.ac.uk

**Data Availability Statement:** Code to reproduce the analyses together with all necessary information is available from our public Github

## Abstract

Quarantine has been long used as a public health response to emerging infectious diseases, particularly at the onset of an epidemic when the infected proportion of a population remains identifiable and logistically tractable. In theory, the same logic should apply to low-incidence infections; however, the application and impact of quarantine in low prevalence settings appears less common and lacks a formal analysis. Here, we present a quantitative framework using a series of progressively more biologically realistic models of canine rabies in domestic dogs and from dogs to humans, a suitable example system to characterize dynamical changes under varying levels of dog quarantine. We explicitly incorporate health-seeking behaviour data to inform the modelling of contact-tracing and exclusion of rabies suspect and probable dogs that can be identified through bite-histories of patients presenting at anti-rabies clinics. We find that a temporary quarantine of rabies suspect and probable dogs provides a powerful tool to curtail rabies transmission, especially in settings where optimal vaccination coverage is yet to be achieved, providing a critical stopgap to reduce the number of human and animal deaths due to rabid bites. We conclude that whilst comprehensive measures including sensitive surveillance and large-scale vaccination of dogs will be required to achieve disease elimination and sustained freedom given the persistent risk of rabies re-introductions, quarantine offers a low-cost community driven solution to intersectoral health burden.

## Author summary

Canine rabies remains a human health risk in many countries around the world, particularly in lower and middle income settings where many dogs are free roaming and able to interact more easily with other dogs and humans. In this paper, we present results from a mathematical model that simulates the spread of rabies both between dogs and from dogs to humans and investigate the impact of quarantine and vaccination at reducing transmission. Our work demonstrates the effectiveness of quarantining both infectious and exposed dogs—we observe that quarantine can have a substantial effect on reducing the number of new animals subsequently infected and thereby lowering the risk of humans

repository https://github.com/IstyRsquared/Quarantine-Models.git.

**Funding:** This work was supported by the Biotechnology and Biological Sciences Research Council the Engineering and Physical Sciences Research Council (BBSRC BB/M01116X/1) granted to KR as part of her doctoral scholarship and by EPSRC GCRF EP/R512916/1 granted to MJT and KR). The funders had no role in study design, data collection and analysis, decision to publish, or preparation of the manuscript.

**Competing interests:** The authors have declared that no competing interests exist.

being exposed to infection. Such a policy can have significant benefits, particularly in settings where access to vaccinations is challenging and resources are limited. Our research can therefore help to inform policy makers in countries where canine rabies is circulating to develop appropriate strategies to reduce the human health risks associated with canine rabies in the future.

## Introduction

Canine rabies, an acute zoonotic infection, has been long an enigma in the field of quantitative epidemiology. While deceivingly easy to trace and hence parameterize as transmission happens predominantly among domestic dogs through saliva of an infected individual, model-based predictions are scarcely ever consistent with empirical observations [1]. It is likely due to the complexity and many interdependent and often unobserved factors of canine rabies ecology that the traditional epidemiological models fail to translate to real-world dynamics in their entirety leading to unrealistic parameter and/or incidence estimates. The details of how rabies transmission operates across temporal and demographic scales has only recently begun to be formally characterized by Mancy et al., 2022 [2]; however, the broader aspects of the disease epidemiology have been widely explored.

International organizations committed to the global elimination of human deaths from dog-mediated rabies by 2030, and scientific guidance to facilitate progress towards elimination has been underway [3, 4]. Decades of operational experience supported by a mounting body of analytical work conclusively demonstrate that mass vaccination of the dog population is the single most important and cost-effective way to control rabies [5–7]. While implementation of high coverage-achieving, spatially comprehensive annual mass dog vaccination campaigns should be prioritized where possible, the desired control efforts may be impeded by logistical constraints such as availability of resources and limited manpower. Supplementary measures to support vaccination campaigns where coverage (temporarily) falls below the recommended threshold (<70% in [8]) are, however, sparse, and often focused on culling of dogs that has been repeatedly shown ineffective in the case of rabies [6, 9, 10].

While immunization of the susceptible population is the primary intervention strategy in the modern world, quarantine—understood as an isolation of confirmed or probable infected cases—is one of the oldest, low-technology forms of disease control [11, 12]. Transmission potential of an infectious disease is driven by the basic reproduction number ($R_0$), defined as the average number of secondary cases caused by an infectious individual in a naïve (non immunized) population. $R_0$ depends on the probability of infection given contact between an infectious and susceptible individual, the length of infectious period, and the number of contacts an infectious individual has per unit time [13]. Both vaccination and quarantine operate by lowering transmission potential through reducing the number of disease-exposure contacts among hosts. Vaccination focuses on the reduction of susceptible individuals available to infection and is particularly of interest with regards to highly transmissible diseases for which a large proportion of a population would be exposed to the disease agent [14–16]. For infections that circulate endemically at low prevalence, or infections at the early stages of an outbreak, contact-tracing followed by quarantine of probable/infectious individuals provides a highly sensitive tool to curtail the transmission potential of a disease [17, 18]. Classic examples of quarantine measures taken at the onset of an epidemic can be found for outbreaks as old as the bubonic plague pandemic in European port cities and early outbreaks of cholera [19, 20], the 1918 pandemic of influenza [21], to more recent emergencies of Ebola [22], the 2009 A

(H1N1)pdm09 influenza [23] and COVID-19 [24]. Conversely, in the case of less-frequent/ low-prevalence infections quarantine is commonly applied as a community-based measure especially in low- and middle-income settings. However, it is sparsely implemented as a public health response, possibly due to the societal and psychological implications of isolation concerning human infections (that outweigh the potential benefits of quarantine in endemic settings) and the lack of formal evidence of its effect for many zoonoses.

Here we seek to develop a set of epidemiological models for rabies transmission to examine the effects of quarantine on the disease dynamics. We first focus on the development of an analytical model to explore the qualitative impact of quarantine on the long-term behaviour of the system, focusing on analysing the theoretical underpinnings of disease persistence and extinction for low prevalence diseases. Building upon the conceptual understanding gained through the mathematical model, we then expand the existing baseline framework by incorporating probabilistic features relevant to rabies ecology. This allows us to quantitatively investigate the changes in rabies dynamics across varying levels of dog vaccination coverage and under the following quarantine scenarios: (1) no quarantine, (2) quarantine of dogs identified through bite-histories of patients presenting at anti-rabies clinics, and (3) enhanced quarantine informed by contact-tracing of rabies suspect and probable dogs.

## Materials and methods

### Theoretical model

In order to investigate the long term dynamics of canine rabies within the dog population, we firstly develop a theoretical model that we can utilise to explore stability properties of the system subject to different values of the basic reproduction number ($R_0$), quarantine rates and vaccination coverage. We consider here an SEIQV model, whereby dogs are either susceptible to infection ($S$), exposed (infected but not yet infectious, $E$), infectious ($I$), quarantined (infectious dogs that are placed in isolation and cannot infect other dogs for the duration of quarantine, $Q$) and vaccinated ($V$). Note that for rabies we assume that all infected individuals subsequently die from disease. We therefore do not explicitly consider the removed class for this model. The equations governing this system can be defined as follows:

$$f(S, E, I, Q, V) = \frac{dS}{dt} = bN - dS - \delta NS - \frac{\beta SI}{N} - v_c S + w_n V \tag{1}$$

$$g(S, E, I, Q, V) = \frac{dE}{dt} = \frac{\beta SI}{N} - dE - \delta NE - \sigma E \tag{2}$$

$$h(S, E, I, Q, V) = \frac{dI}{dt} = \sigma E - (d + \delta N + \gamma + q)I \tag{3}$$

$$i(S, E, I, Q, V) = \frac{dQ}{dt} = qI - (d + \delta N + \tau)Q \tag{4}$$

$$j(S, E, I, Q, V) = \frac{dV}{dt} = v_c S - (d + \delta N + w_n)V \tag{5}$$

where $N = S + E + I + Q + V$. In this set of equations, $b$ is the birth rate, $d$ is the natural death rate, $\sigma$ is the rate of transition from the exposed to the infectious class, $\gamma$ is the death rate from disease, $v_c$ is the vaccination rate, $w_n$ is the rate of waning immunity following vaccination and $\tau$ is the rate of removal from quarantine. We assume that density dependent natural deaths of

dogs occur at rate $\delta N$ where $\delta = \frac{b-d}{K}$ and $K$ is the carrying capacity of the population in line with previous work by [25]. In contrast to the computational model below in which vaccinated dogs can be re-vaccinated before their immunity wanes, in the deterministic framework only susceptible dogs can be vaccinated.

At epidemic onset and in the absence of quarantine ($q = 0$), we can therefore define the basic reproduction number for this system as

$$R_0 = \frac{\sigma}{d + \delta N + \sigma} \times \frac{\beta}{d + \gamma + \delta N}$$

where $\frac{\sigma}{d+\delta N+\sigma}$ is the fraction of individuals who successfully progress from the exposed to the infectious class and $\frac{\beta}{d+\gamma+\delta N}$ is the transmission rate divided by the average duration that an individual is infectious for.

We will now explore the stability of the system as it approaches equilibrium. Equilibrium solutions occur when $\frac{dS}{dt} = \frac{dE}{dt} = \frac{dI}{dt} = \frac{dQ}{dt} = \frac{dV}{dt} = 0$. From Eqs 3 and 4, in equilibrium we find that

$$E^* = \frac{(d + \delta N + \gamma + q)I^*}{\sigma} \tag{6}$$

$$Q^* = \frac{qI^*}{d + \delta N + \tau} \tag{7}$$

We now substitute our expression for $E^*$ in Eq 6 into Eq 2 such that, in the endemic equilibrium (when $\frac{dE}{dt} = 0$ and $I \neq 0$) we find:

$$S^* = \frac{N(d + \delta N + \sigma)(d + \delta N + \gamma + q)}{\beta \sigma} \tag{8}$$

Substituting this expression into Eq 5, we can obtain an expression for $V^*$ in the endemic equilibrium:

$$V^* = \frac{v_c N(d + \delta N + \sigma)(d + \delta N + \gamma + q)}{\beta \sigma (d + \delta N + w_n)} \tag{9}$$

We can now substitute our expressions for $S^*$ and $V^*$ into Eq 1 to obtain an expression for $I^*$:

$$I^* = \frac{bN\sigma}{(d + \delta N + \sigma)(d + \delta N + \gamma + q)} - \frac{N(d + \delta N + v_c)}{\beta} + \frac{w_n N v_c}{\beta(d + \delta N + w_n)} \tag{10}$$

Finally, we can substitute the expression for $I^*$ into Eqs 6 and 7 to obtain expressions for $E^*$ and $Q^*$:

$$E^* = \frac{bN}{d + \delta N + \sigma} - \frac{N(d + \delta N + v_c)(d + \delta N + \gamma + q)}{\beta \sigma} + \frac{w_n N v_c (d + \delta N + \gamma + q)}{\beta \sigma (d + \delta N + w_n)}$$ (11)

$$Q^* = \frac{q}{d + \delta N + \tau} \left( \frac{bN\sigma}{(d + \delta N + \sigma)(d + \delta N + \gamma + q)} - \frac{N(d + \delta N + v_c)}{\beta} + \frac{w_n N v_c}{\beta(d + \delta N + w_n)} \right)$$ (12)

In order to analyse the stability of the system, we need to calculate the Jacobian matrix, $J$. Given that $N = S + E + I + Q + V$ and assuming that $N$ is fixed (such that the natural birth rate compensates for natural death and death as a result of infection), we can set $V = N - S - E - I - Q$ and reduce the system to consider the four variables $S$, $E$, $I$ and $Q$. $J$ for this system is

$$J = \begin{bmatrix} \frac{\partial f}{\partial S} & \frac{\partial f}{\partial E} & \frac{\partial f}{\partial I} & \frac{\partial f}{\partial Q} \\ \frac{\partial g}{\partial S} & \frac{\partial g}{\partial E} & \frac{\partial g}{\partial I} & \frac{\partial g}{\partial Q} \\ \frac{\partial h}{\partial S} & \frac{\partial h}{\partial E} & \frac{\partial h}{\partial I} & \frac{\partial h}{\partial Q} \\ \frac{\partial i}{\partial S} & \frac{\partial i}{\partial E} & \frac{\partial i}{\partial I} & \frac{\partial i}{\partial Q} \end{bmatrix}$$

$$= \begin{bmatrix} -d - \delta N - \frac{\beta I}{N} - v_c - w_n & -w_n & -w_n & -w_n \\ \frac{\beta I}{N} & -d - \delta N - \sigma & \frac{\beta S}{N} & 0 \\ 0 & \sigma & -d - \delta N - \gamma - q & 0 \\ 0 & 0 & q & -d - \delta N - \tau \end{bmatrix}$$

To determine the behaviour of this system, we need to calculate the eigenvalues of the Jacobian. We therefore need to find the solution of $|J - \lambda I| = 0$ such that:

$$\begin{vmatrix} -d - \delta N - \frac{\beta I}{N} - v_c - w_n - \lambda & -w_n & -w_n & -w_n \\ \frac{\beta I}{N} & -d - \delta N - \sigma - \lambda & \frac{\beta S}{N} & 0 \\ 0 & \sigma & -d - \delta N - \gamma - q - \lambda & 0 \\ 0 & 0 & q & -d - \delta N - \tau - \lambda \end{vmatrix} = 0$$

To evaluate the stability of the endemic equilibrium in the context of intervention strategies that might be applied to the system, we use a numerical solving method to evaluate the Jacobian at the endemic equilibrium and calculate the eigenvalues for given values of $R_0$, vaccination rate $v_c$ and quarantine rate $q$. We determine both when the endemic equilibrium is stable and how the number of infected individuals at the endemic equilibrium depends upon these quantities. A flow diagram of the single-species model is shown in Fig 1. All computational work is performed in the programming environment R, version 4.3.0.

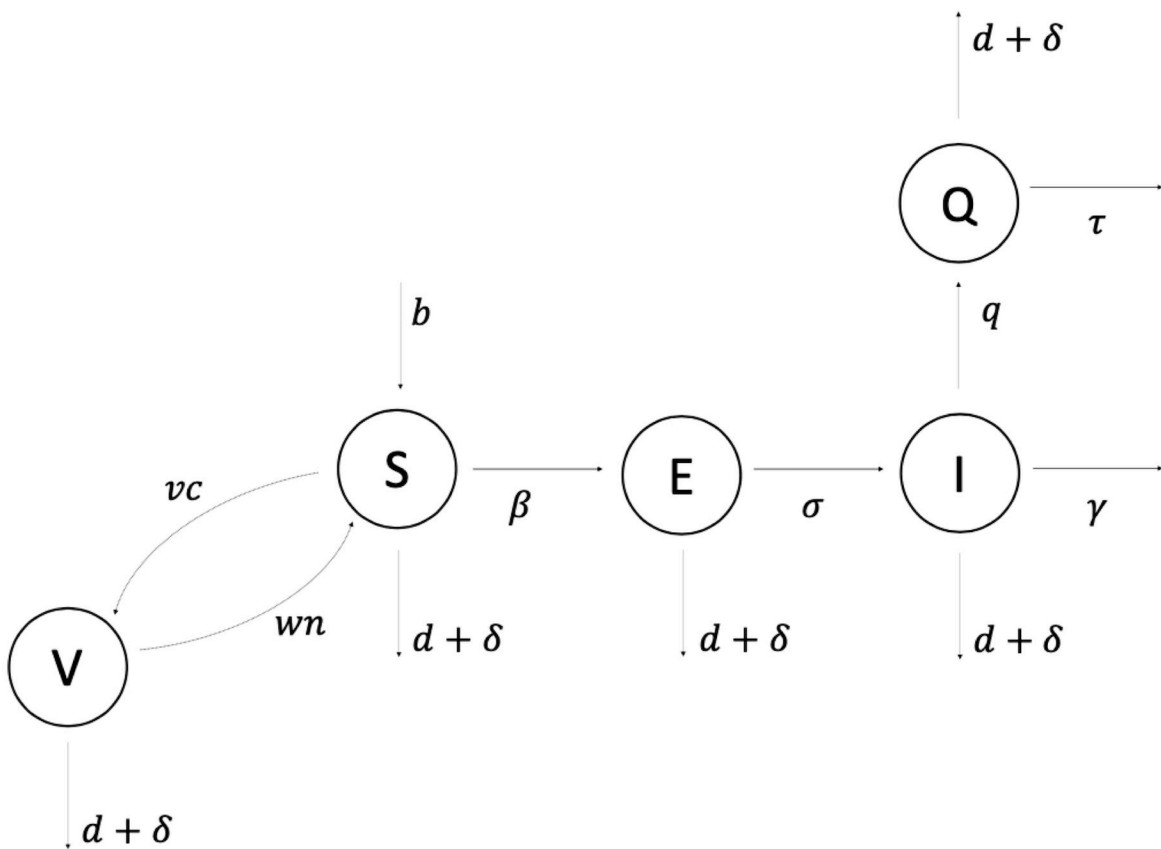

**Fig 1. Single-species SEIQV deterministic compartmental model diagram.** Epidemiological classes are indicated by circles and arrows suggest the directionality of transitional flows of individuals and the virus moving between compartments. Susceptible individuals can become either Exposed at rate $\beta$ or Vaccinated at rate $v_c$. Vaccinated individuals then return to the Susceptible class with waning immunity of the vaccine at rate $w_n$ which is given by the reciprocal of the average longevity of the vaccine. Infectious individuals can be taken out of their class and placed into Quarantine at rate $q$ (note that here we assumed that only Infectious individuals can transition into the Quarantined class). In this scheme, all infectious dogs are removed from the population and die at rate $\gamma$ for Infected individuals and rate $\tau$ for the Quarantined dogs. Here $\tau = \gamma/(1 - \gamma\rho)$ where $\rho$ is the mean delay from an individual becoming infectious to entering quarantine, assuming all Quarantined individuals will always die and be removed from the population.

## Computational model

Whilst the theoretical model presented above provides useful insights regarding the evolution of a rabies-like system in the presence of vaccination and quarantine, transmission of the virus in real-world settings is highly stochastic and can be significantly influenced by low probability events, such as incursions of infected animals into the population or super-spreading events. Stability analyses of deterministic models ignore the role of stochasticity with regards to the pathogen extinction and reintroduction (both locally and globally) despite its influential impact on the future trajectories for diseases that operate at low transmission levels. For example, the probability of a disease going extinct decreases with an increasing value of $R_0$ and vice versa [26]. As such, the deterministic threshold for elimination will be modulated by chance processes that can break individual chains of transmission resulting in a faster elimination, or allow the pathogen to persist for longer through a series of infection events and chance re-introductions in spite of an overall high level of immunity within the population. In the case of the analytical model, we also ignore the variability in the duration of exposure which spans a wide temporal range. Symptoms of rabies in dogs usually manifest in the first month following

exposure, but the incubation period may last for several months, effectively functioning as an "endogenous" incursion [27, 28]. This becomes particularly relevant when we allow for Exposed dogs that are quarantined but do not show any symptoms of the disease by the end of the quarantine period to re-enter the population and cause further infection despite their previous quarantine status.

We therefore develop a stochastic SEIRQV compartmental model to simulate the spread of disease in the dog population, coupled with an SEIRV model for humans. The system is summarised in Fig 2. In order to introduce a degree of biological realism, we expand the existing SEIQV model by the following additions most relevant from the empirical work. We first (1) introduce the role of stochasticity by modelling both the disease and population dynamics as a probabilistic process, (2) re-define the transmission rate to capture heterogeneity in individual biting behaviour, and (3) allow for exogenous incursions to enter the population. We then introduce an R compartment to keep count of all dogs "Removed" from the population by natural death and the disease (4). We build further realism to modelling the dog quarantine practice, by (5) allowing for a potential removal of Exposed dogs from quarantine when disease symptoms do not occur within the recommended time period of dog exclusion. Lastly, we extend the single host model to (6) include transmission from dogs to humans, and (7) to

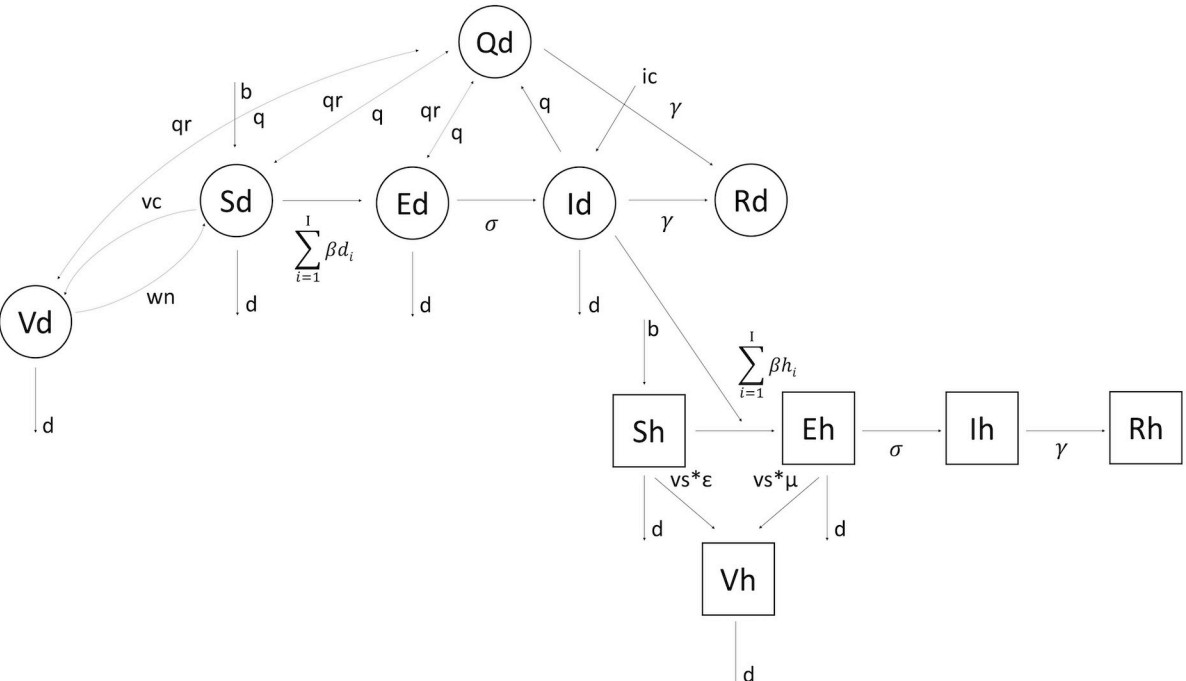

**Fig 2. Multi-species SEIRVQ compartmental model diagram.** Dog and human epidemiological classes are indicated by circles and squares respectively. Arrows show directions at which individuals and the pathogen move through the system. Compared to the single-species model, here we allow for any epidemiological class of the dog population (except for the Removed class) to be placed in quarantine. Susceptible and Vaccinated quarantined dogs are returned to their respective compartments upon completion of the quarantine at the rate $q_r$. Infectious quarantined dogs are removed from the population as a result of disease-induced death at the rate $\gamma$. Depending on the progression of the disease in Exposed quarantined dogs, two distinct scenarios can occur. For those individuals that will become Infectious within the time frame of their quarantine, disease-induced death follows at the same rate as for Infected individuals, whereas Exposed quarantined dogs that are asymptomatic by the end of their quarantine are returned back into the Exposed class at the rate $q_r$. Transmission of the disease between dogs, and from dogs to humans is defined as a sum of offspring rabid bites seeded by Infectious individuals, drawn from a negative binomial distribution taking different parameter values for dogs and human. Lastly, the overall level of infection in the system can be elevated by an introduction of exogenous incursions entering the system at the rate $i_c$. All model parameters associated with the disease and demographic processes illustrated here are summarized in Table 1.

incorporate information on health-seeking behaviour collected through a longitudinal enhanced surveillance study of dog bite-injury patients [29] in order to approximate the probability of quarantine under different surveillance scenarios.

Specifically, we model the time stepping process weekly using the Tau leap algorithm [11, 30, 31]. We then parameterise rabies transmission explicitly as the number of rabid bites per infectious individual. Offspring exposures (here representing a secondary exposure resulting from a biting incident caused by a primary case individual, not a vertical transmission from a parent to its offspring) are drawn from a negative binomial distribution as

$$\beta_i \sim NB(R_0, k)$$

where $R_0$ is the expected number of new infectious bites and $k$ takes different values for humans and dogs. The $k$ value here can be understood as analogous to variance around a mean which tends to be wider for dog-dog transmission (leading to longer tailed distributions) as opposed to dog-human transmission. The total transmission rate $\beta$ at each time step is then formulated as a sum of all offspring exposures present in the system at the modelled time step, multiplied by the probability of a bite becoming a case ($\sim$ 50% in [32]), and distributed proportionally to the size of each compartment available to exposure (all except for individuals in Quarantine).

$$\beta = 0.49 * \sum_{i=1}^{I} \beta_i$$

Incursions $i_c$ are drawn from a Poisson distribution where

$$i_c \sim Poiss(\bar{i})$$

The true probability at which a population receives an incursion will likely vary over time as control is implemented in neighbouring populations, and geographically given the localized heterogeneous nature of rabies incidence. Here, we incorporate incursions to maintain fluidity in the disease system but initially set the value to function only as a "background" probability ($\bar{i} = 1.5$). We then test the role of incursions on rabies dynamics by either removing the incursions entirely or increasing the incursion probability by two fold.

We define dog quarantine as the number of dogs identified through a triage of patients presenting at anti-rabies clinics. The number of quarantined dogs is then drawn from a Conway-Maxwell-Poisson distribution as an extension to a Poisson distribution that allows for modelling both over- and under-dispersion:

$$Q_i \sim CMPoiss(q, range)$$

where $q$ and $range$ differ between investigations of case (understood as patients bitten by a rabid dog) and non-case (understood as patients bitten by a non-rabid dog) incidents. For rabies Exposed and Infected dogs (divided proportionally according to the duration of incubation and infectious periods) identified through patient investigations, the total number of additional dogs per time step moved into quarantine is then calculated as

$$Q = \eta \sum_{i=1}^{V_h} Q_i \tag{13}$$

where $\eta$ is the probability that the dogs identified through following rabid animals responsible for exposed case patients are also infected with rabies (i.e., Exposed or Infected). It is important to note that $\eta$ only captures those dogs identified through following rabid animals responsible

for case patients and therefore is only a fraction of the total dogs that may be exposed or infectious in the population at any time. From field observations we believe that $\eta$ is relatively high [29], but for modelling purposes here we opt for a more conservative assumption of 70% as more data are needed for rigorous estimates. Otherwise, the dogs responsible for non-case incidents both in humans and dogs are distributed in proportion to the size of each relevant compartment (i.e., $S$, $E$, $Q$, and $V$ within the dog population).

As the current recommendations by the Bureau of Animal Industry and the Department of Health in the Philippines are that unvaccinated dogs that bite a known patient should be quarantined and observed for 14 days, we utilise this period in our analysis [33, 34]. In our model, given the duration of the incubation period is longer than the duration of quarantine (22.3 days in [32] and 14 days respectively), a fraction of Exposed quarantined dogs may not become symptomatic before their release. To account for such a possibility, we explicitly generate days until symptomatic for each exposed dog held in quarantine, and return those individuals showing no symptoms after the 14-day period back into the population.

Immunity of humans is defined as achieved through administration of PEP upon attendance at a clinic (note, here we assume that two doses of PEP delivered at days 1 and 7 would provide immunity). The weekly proportion of bite-injury patients $\epsilon$ is drawn from a zero truncated normal distribution (as the number of patients presenting at the clinics follows a normal distribution bounded by zero from below the mean) of weekly throughput records collected at the anti-rabies clinics [29]. The percentage of rabies exposed case patients that will receive PEP ($\mu$) varies extensively across geographical areas and socioeconomic backgrounds. Here, we assume that with enhanced surveillance 80% of human cases would be detected in a timely manner and administered the lifesaving vaccine.

Lastly, we incorporate bias in re-vaccination of dogs ($v_v$) directed towards individuals that are easy to capture for administration of the vaccine. Importantly, the model structure has been developed such that we are able to capture not just rabid dogs that bite other dogs and humans, but also rabid dogs that do not bite as well as potentially exposed humans that do not attend a clinic or receive PEP. We are thus able to provide a close-to-realistic representation of canine rabies dynamics in most low- and middle-income settings for which this modelling study is intended. All parameters are summarized in Table 1.

We utilise our computational model to investigate the impact of varying levels of quarantine on rabies dynamics under four vaccination scenarios: 0%, 25%, 50% and 75% of the dog population. We then test three progressively strengthened quarantine scenarios motivated by an Integrated Bite Case Management study conducted in the Philippines [29]. Here [29] collected information on bite-histories of patients presenting at anti-rabies clinics through a series of interviews and phone follow-ups combined with field investigations to determine the epidemiological status of biting animals and to provide quarantine recommendations. In the model, no quarantine is implemented under Scenario 1. In Scenario 2 we assume only dogs identified through bite-injury patients presenting at anti-rabies clinics would be quarantined, suggesting a medium-level quarantine (understood as a smaller proportion of the dog population ending in quarantine, rather than a less-strict practice) with an average number of dogs per patient (both non-case, and case patients) around 1 (drawn from Conway-Maxwell-Poisson distribution where $q = 2.5$ and *range* = 4.3). Under Scenario 3 we assume a triage of bite-injury patients as per Scenario 2, but this time coupled with further field investigations and contact tracing of patient biting dogs. In scenario 3 we expect the number of dogs identified for quarantine through non-case patients to remain within the same range as in Scenario 2 but to increase for investigations informed by case patient incidents (drawn from Conway-Maxwell-Poisson distribution again where $q = 2.8$ and *range* = 1.5). A computed distribution of the number of dogs identified for quarantine under each of the tested treatments is shown in Fig 3. Lastly, we

**Table 1. Summary of SEIRVQ model parameters.** Parameter values are provided for dogs and humans separately with respective source references.

| Parameter | Description | Value (dogs) | Value (humans) | Source (dogs) | Source (humans) |
|---|---|---|---|---|---|
| $R_0$ | Basic reproduction number | 1.3 | 0.37 | [6] | [2] |
| $k$ | Clumping parameter | 1.33 | 0.56 | [6] | [44] |
| $\bar{i}$ | Mean number of introduced cases | 1.5 | - | - | - |
| $\sigma$ | 1/incubation period | 1/22.3 days | 1/40 days | [32] | Warrell in [27] |
| $\gamma$ | 1/infectious period | 1/3.1 days | 1/7 days | [32] | Warrell in [27] |
| $b$ | Per capita birth rate | 0.38 per year | 0.019 per year | [7] | [52] |
| $d$ | Per capita death rate | 0.28 per year | 0.0052 per year | Estimated from data | [52] |
| $\frac{v_c}{v_s}$ | Vaccination coverage/rate | 0%, 25%, 50% and 75% | 1/8 days | - | [29] |
| $v_v$ | Vaccination bias | 0.4 | - | - | - |
| $w_n$ | 1/Duration of vaccine | 0.33 per year | | [53] | - |
| | 1/Duration of vaccine | | | | |
| $q_r$ | 1/Duration of dog quarantine | 1/14 days | - | - | - |
| $\mu$ | Probability of receiving PEP after rabies exposure | - | 0.8 | - | - |
| $\epsilon$ | Proportion of non-case patients presenting at clinic | - | Drawn from a distribution of weekly throughput | | Truncated normal distribution |
| $\eta$ | Probability of identifying another rabies infected dog through follow up of rabies exposed patients | 0.7 | - | - | - |

observe rabies dynamics under three incursion scenarios by setting the incursion probability to $\bar{i} = 0$, 1.5 (default) and 3. Given the highly stochastic nature of the model, each scenario is iterated 1000 times, run over 10 consecutive years and explored across the same range of $R_0$ values as in the analytical model above (spanning from 1 to 2 in 0.1 sized increments).

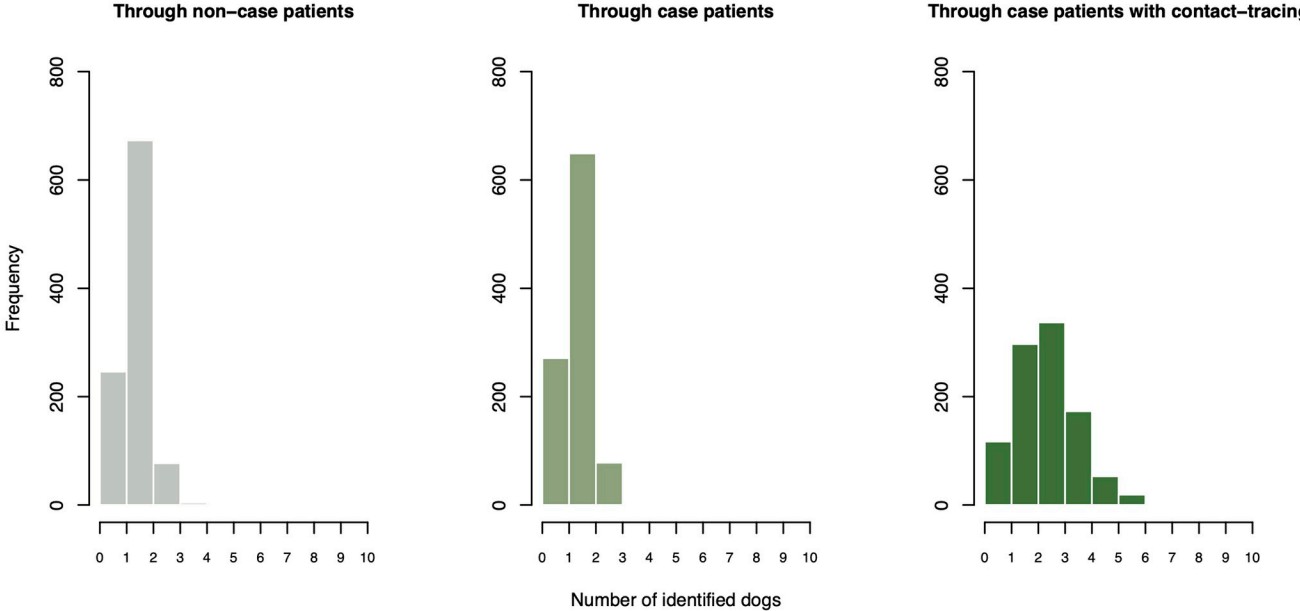

**Fig 3. Distributions of the number of dogs identified for quarantine across surveillance scenarios.** Frequency distribution of the number of dogs identified for quarantine per biting dog responsible for (from left to right) non-case patients, rabies exposed case patients, and rabies exposed case patients coupled with additional in-field contact tracing investigations of these incidents. Estimates are drawn from a Conway-Maxwell-Poisson distribution with different parameter values taken for each scenario.

## Results

### Stability analysis of theoretical model

Existing models fitted to rabies time-series data suggest that the distribution of $R_0$ falls predominantly between 1 and 2 [6, 32, 35, 36]. As such, we vary the transmission rate $\beta$ by gradually increasing the value of $R_0$ from 1 to 2 in increments of 0.1 (where $q = 0$ initially to emulate the baseline transmission rate under no intervention). We then set the percentage of Infectious dogs terminating in quarantine every week ($q_P$) to vary between 0 and 100% in 5% increments, where the rate of quarantine $q = -\frac{\ln(1-q_p/52)}{\Delta t}$ and $\Delta t = 1$ (note, all parameters are expressed as weekly rates). To explore the stability of the endemic equilibrium dependent upon a given vaccination coverage, quarantine rate and value of $R_0$, we also vary the mean percentage of dogs vaccinated per year, $v_p$ in 5% increments from 0% to 100%. The weekly vaccination rate is calculated in terms of the percentage of vaccinated dogs such that $v_c = -\frac{\ln(1-v_p/52)}{\Delta t}$ where $\Delta t = 1$. All remaining parameters used in both models are summarized in Table 1, except for the rate at which Infectious dogs leave the quarantine class, which is defined as $\tau = \gamma/(1 - \gamma\rho)$, where $\rho$ is the mean delay from an individual becoming infectious to entering quarantine.

For each combination of parameter values, we then calculate representative eigenvalues to determine the stability of the endemic equilibrium and the size of the infected population (i.e., the number of Exposed and Infectious dogs) for parameter combinations at which the endemic equilibrium is found to be stable. The results are summarised in Fig 4.

Initially we consider the case of $R_0 = 1.3$, which previous research indicates as the most likely value for the basic reproduction number for rabies [2]. In the absence of quarantine, we find that as vaccination rates approach 0.35 the endemic equilibrium loses stability and the disease free equilibrium becomes stable. Similarly, in the absence of vaccination, the same result occurs for quarantine rates above 0.45 (Fig 4A). As the levels of vaccination increase, lower levels of quarantine are required to result in the endemic equilibrium losing stability and the virus being eliminated. As we approach this transition we note that the number of both Infectious and Exposed dogs in the endemic equilibrium decreases, highlighting the effectiveness of vaccination and quarantine at reducing the overall transmission in the population (Fig 4B and 4C). However, to maintain disease endemicity for $R_0 = 1.3$ in the absence of both interventions, the model predicts unrealistically high incidence of the disease ($> 135$ infected dogs per 10,000) exceeding what is suggested by empirical evidence.

We now explore the impact of different values of $R_0$ upon the stability properties of the endemic equilibrium as vaccination and quarantine rates are varied. When $R_0$ is close to 1 only very low rates of vaccination and/or quarantine are required in order for elimination to occur. As $R_0$ increases towards 2, much higher quarantine and vaccination resources are required in order for the endemic equilibrium to become unstable. When $R_0 = 2$ (which we note represents the upper limit of a realistic value for the basic reproductive number for rabies) we observe that it is possible for rabies to be eliminated provided that sufficient rates of quarantine and vaccination are maintained (Fig 4B and 4C). However, the increased rates of vaccination and particularly quarantine required for elimination in this scenario may be unrealistic and/or infeasible in practice given constrained resources for vaccination, the limited capacity to successfully identify infected dogs for quarantine and the ability to isolate a large number of dogs at any given time. In particular, the levels of quarantine required for elimination in the absence of vaccination would not be possible in practice in real world settings, which emphasises the critical role vaccination plays in achieving elimination of rabies.

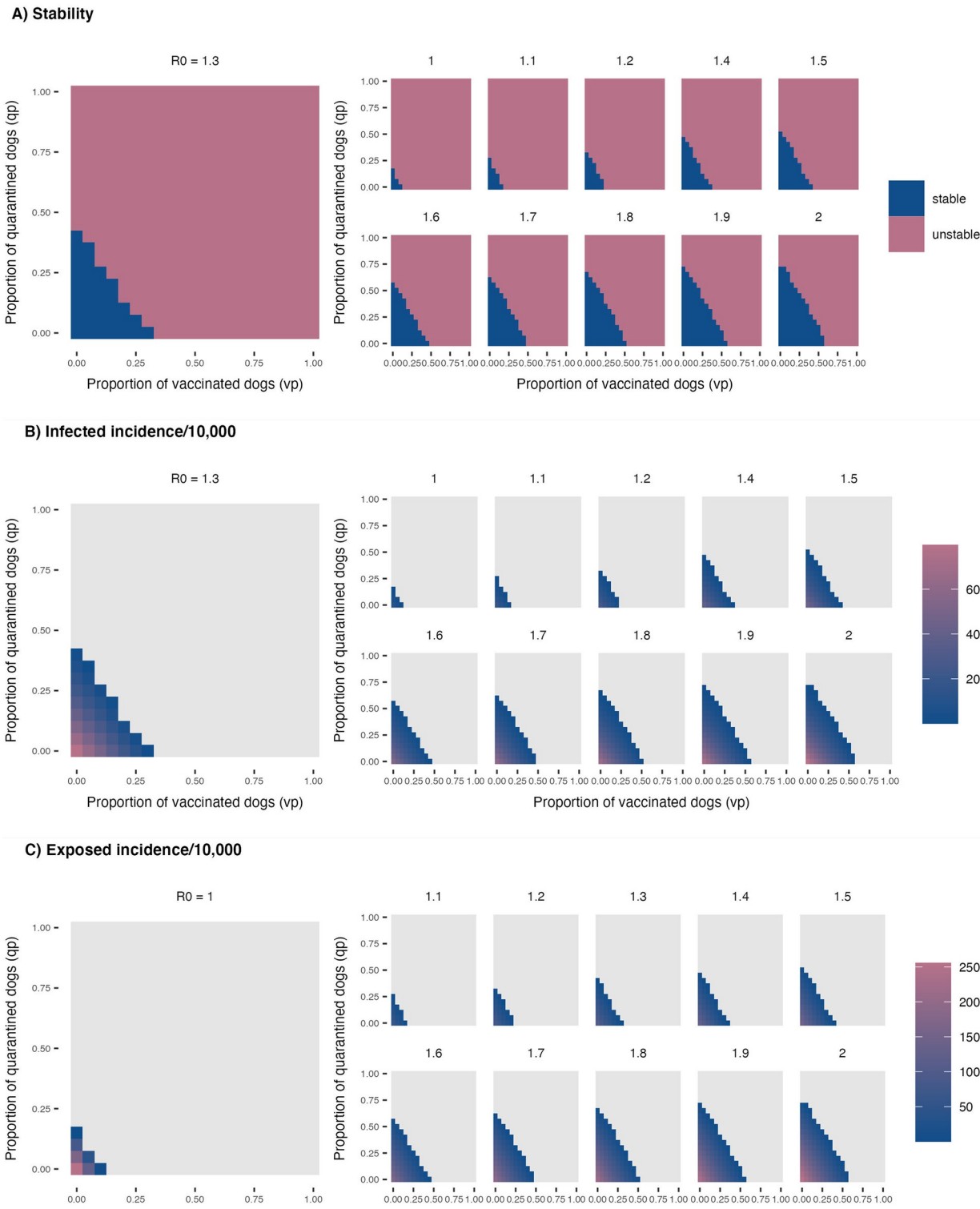

**Fig 4. Rabies dynamics across vaccination and quarantine parameter space for different values of $R_0$. A**: Stability analysis for the deterministic SEIQV model under $R_0$ values ranging from 1 to 2. Raster shading shows the stability at the endemic equilibrium for different combinations of vaccination and quarantine rates. The blue shading indicates the region where the endemic equilibrium is stable, whilst the red region indicates where the endemic equilibrium is unstable. **B**: Case incidence of infected dogs per 10,000 for vaccination and quarantine rates from 0 to 1 when the system is in endemic equilibrium. In this panel grey shading indicates the region where the disease free equilibrium is stable. **C**: Case incidence of exposed dogs per 10,000 for vaccination and quarantine rates equivalent to values shown in B.

## Computational analysis

Any deterministic framing precludes variability in parameter values and the role of chance. To address the key limitations of the deterministic framework, we build a stochastic discrete-time multispecies SEIRVQ model with explicit individual biting behaviour and the probability of rabies incursions entering the system from outside. Here we observe the impact of chance events on the disease dynamics under a varying degree of dog vaccination and quarantine, and increasing incursion probabilities.

In line with the previous results obtained from the deterministic setting, increasing quarantine and vaccination coverage has a positive effect on curtailing the epidemic and leads to significant reductions in the overall number of infections, both in humans and dogs (Fig 5 and Table 2). However, introductions of rabies cases from outside the population pose an additional impediment to disease elimination. Exogenous incursions increase the magnitude of transmission temporarily and decrease the probability of extinction in the long term. In the stochastic formulation, neither the vaccination nor the quarantine interventions result in a complete interruption of transmission when the incursion probability $>= 1.5$. In fact, under intensified vaccination efforts and with increased detection and quarantine of infected dogs, endemicity appears to be sustained predominantly through incursions (Figs 6 and 7).

Similar dynamics have been reported for diseases with lower transmission rates [37] and/or during the endgame (i.e. pre-elimination/pre-eradication epidemiological stage as described in [38, 39]), when heterogeneities in the force of infection (often driven by incursions) and the level of immunity may result in unpredictable stochastic outbreaks [38, 40]. Conversely, in the absence of incursions, rabies transmission appears to die out regardless of the vaccination coverage and for $R_0 < 2$ (Fig 6), albeit the timelines to elimination decrease with increasing control efforts. For $R_0 = 2$ rabies transmission persists only in no vaccination settings upon removal of exogenous incursions.

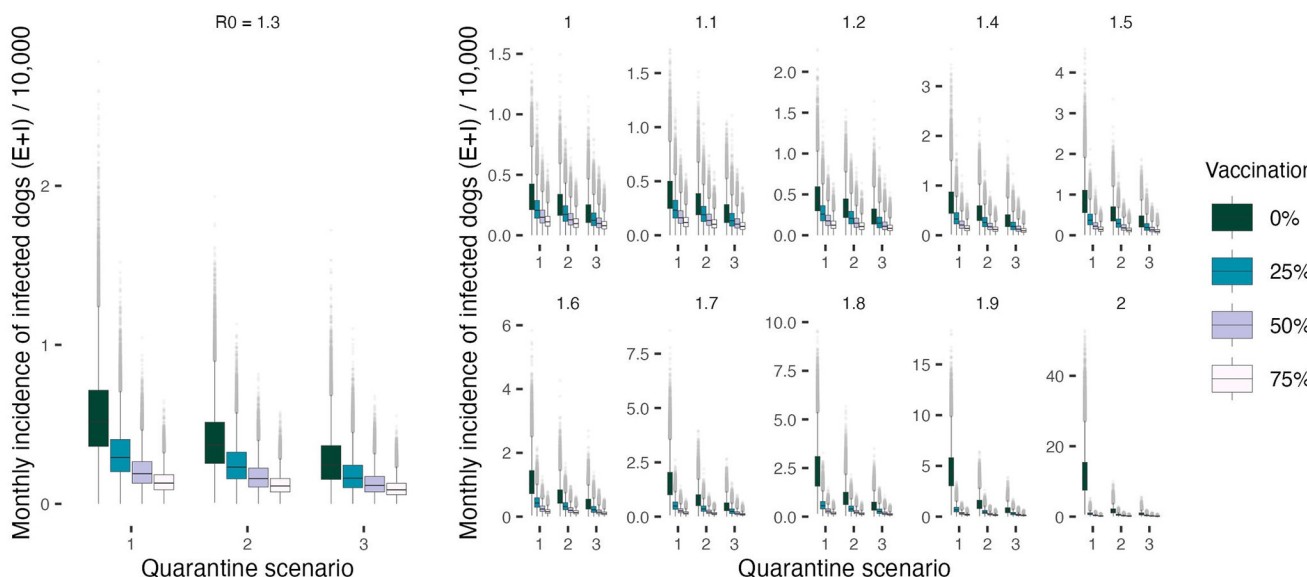

**Fig 5. Incremental decline in the case incidence (Exposed and Infected dog cases) per 10,000 dogs across 1000 simulations with increasing levels of dog quarantine and vaccination coverage.** Summaries shown as individual panels for a range of $R_0$ values (note the differential scale of y-axes between the panels). Differences between quarantine scenarios in the number of Exposed and Infectious dogs is particularly striking in low vaccination coverage settings and/or for higher values of $R_0$ as supported by the statistical analysis summarized in Table 2. The simulation data points do not include the initial burn-in period of the first six months.

**Table 2. Trends in the weekly number of infected dogs (Exposed and Infectious) (Top table) and human deaths due to rabid bites (Bottom table) under vaccination and quarantine scenarios.** All variables statistically significant with $p \ll 0.005$. Incident rates for vaccination (continuous) and quarantine (categorical) treatments are calculated from regression coefficients obtained via a negative binomial general linear model fitted to the simulated time series. The incident rates indicate a multiplicative effect on the unit change in the response variable. For example, for $R_0 = 1.3$ the expected number of canine rabies cases is 0.79 times the reference value (intercept) when Quarantine Scenario 2 is implemented as opposed to no quarantine, and 0.59 times the reference value when Quarantine Scenario 3 is implemented compared to no quarantine (i.e., reducing the expected mean of 55.87 to 44.58 and 33.13 for Scenarios 2 and 3 respectively). For the same value of $R_0$, every unit increase in vaccination efforts (i.e., 1%) would change the expected mean of canine cases by 0.2%, giving to the expected mean of 25.23 infected dogs for 50% vaccination coverage. Whilst the impact of quarantine on human health is only indirect through the reduced number of rabid dogs circulating in the population and the weekly incident rates of the quarantine scenarios relatively low, on average three humans lives would be saved every month when either quarantine scenario is implemented assuming the same basic reproductive number as above.

| $R_0$ | Canine Infections Intercept | Vaccination Incident Rate | Quarantine Scenario 2 Incident Rate | Quarantine Scenario 3 Incident Rate |
|---|---|---|---|---|
| 1.0 | 34.82 | 0.302 | 0.834 | 0.648 |
| 1.1 | 40.47 | 0.268 | 0.825 | 0.628 |
| 1.2 | 47.37 | 0.235 | 0.815 | 0.613 |
| 1.3 | 55.87 | 0.204 | 0.798 | 0.593 |
| 1.4 | 66.25 | 0.174 | 0.779 | 0.571 |
| 1.5 | 80.48 | 0.147 | 0.756 | 0.543 |
| 1.6 | 99.63 | 0.120 | 0.732 | 0.521 |
| 1.7 | 128.66 | 0.093 | 0.687 | 0.480 |
| 1.8 | 175.71 | 0.069 | 0.628 | 0.431 |
| 1.9 | 277.10 | 0.045 | 0.535 | 0.359 |
| 2.0 | 579.98 | 0.023 | 0.380 | 0.251 |
| $R_0$ | Human Deaths Intercept | Vaccination Incident Rate | Quarantine Scenario 2 Incident Rate | Quarantine Scenario 3 Incident Rate |
| 1.0 | 0.91 | 0.605 | 0.875 | 0.797 |
| 1.1 | 0.99 | 0.563 | 0.858 | 0.772 |
| 1.2 | 1.09 | 0.510 | 0.846 | 0.753 |
| 1.3 | 1.21 | 0.462 | 0.832 | 0.729 |
| 1.4 | 1.36 | 0.409 | 0.813 | 0.704 |
| 1.5 | 1.55 | 0.351 | 0.785 | 0.668 |
| 1.6 | 1.82 | 0.299 | 0.752 | 0.638 |
| 1.7 | 2.24 | 0.232 | 0.699 | 0.577 |
| 1.8 | 2.98 | 0.171 | 0.633 | 0.505 |
| 1.9 | 4.63 | 0.103 | 0.519 | 0.404 |
| 2.0 | 9.49 | 0.046 | 0.361 | 0.265 |

To further probe the interaction between vaccination and quarantine measures, we test four incrementally increasing levels of vaccination coverage (i.e., 0%, 25%, 50% and 75%. Whilst for the deterministic framework, $> 20\%$ vaccination coverage was found to be sufficient in order to drive the system to extinction in the presence of low-level quarantine (Fig 4A), this threshold is likely inaccurate for a system receiving infected cases form outside, with fast turnover of susceptible individuals through high birth and death rates, and variability in the number of offspring cases for each Infectious dog. The effects of quarantine (both medium and high level—scenarios 2 and 3 respectively) on the number of infections in the population as well as human deaths appear particularly significant in no- and low-vaccination settings, with less apparent yet statistically significant impact as the vaccination coverage increases (Fig 5 and Table 2).

## Discussion

Canine rabies circulating in domestic dogs represents a serious burden on public health budgets and local communities. Mass dog vaccination campaigns, the cornerstone of effective

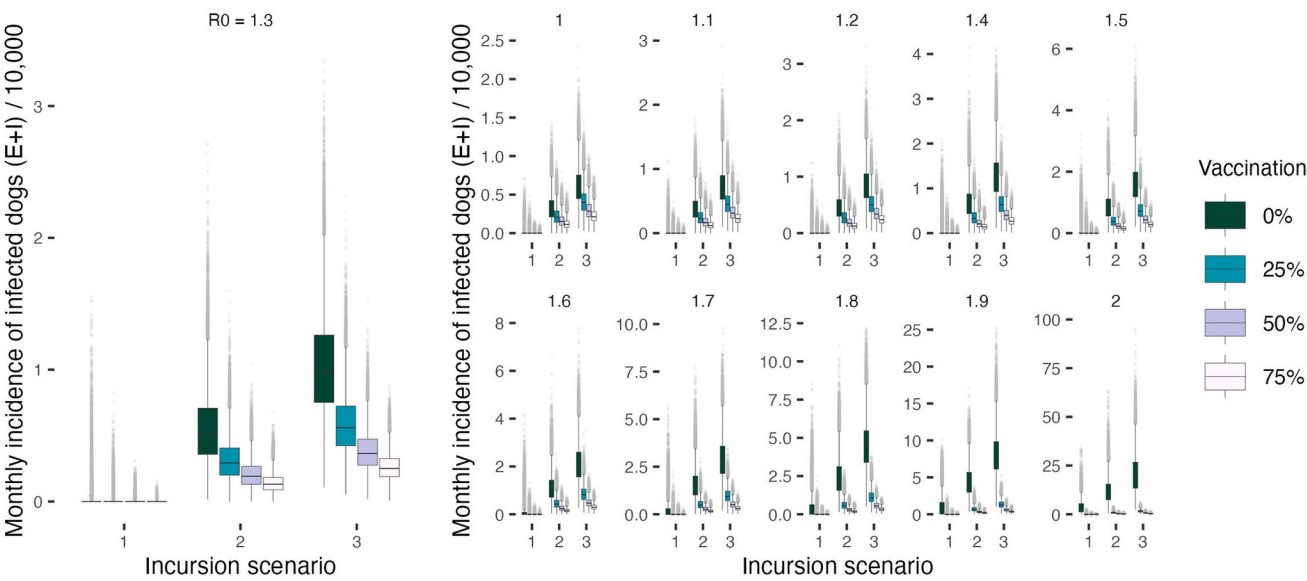

**Fig 6. Summary of case incidence (Exposed and Infected dog cases) per 10,000 dogs across 1000 simulations assuming increasing incursion probability and vaccination coverage values.** Summaries organized as in Fig 5. Incursion scenarios 1, 2 and 3 indicate an incursion probability $\bar{i}$ equal to 0, 1.5, and 3 respectively. In the absence of exogenous incursions, rabies transmission in the modelled population eventually dies out regardless of the vaccination coverage for all values of $R_0$ expect for $R_0 = 2$. The time to elimination will, however, decrease with intensified vaccination efforts. The simulation data points again do not include the initial burn-in period of the first six months.

rabies control, has led to elimination of human deaths and interruption of rabies transmission at the source in most countries across the Global North [41]. Such campaigns, however, require systematic efforts delivered at scale and sustained over long periods of time [6]. As the availability of human and financial resources is typically limited in low- and middle-income countries, questions remain over the most effective strategies to control rabies given extensive technical and structural constraints.

Contact tracing and subsequent quarantine of infected individuals play an important role in the control of infectious diseases at the onset of an outbreak or during the endgame [42, 43]. Concentrating control on infected contacts can be potentially extremely effective, but it relies on a sensitive surveillance system and a logistically traceable fraction of the infected population. Such conditions are typically met for diseases with easily recognizable symptoms and during the early and/or final stages of an epidemic or for endemic pathogens persisting at low prevalence when only a limited number of infections is present within the system.

Whilst largely endemic, rabies provides a suitable system to test a wider use of quarantine outside its traditional application. Biting behaviour is the primary indicator of rabies; transmission events, particularly from dogs to humans, are extremely memorable as often inducing severe distress or even psychological trauma. Thus, they are relatively easy to identify (when investigated) and traced back and forward as the local communities remember the bite histories long after they have occurred. In addition, rabies circulates at a low prevalence with $R_0 < 2$, indicating only a small percentage of the population is being infected at any time [32, 44].

The concept of quarantine for rabies suspect and probable dogs has long been part of the general guidelines for community-based rabies control, particularly in low- and middle-income countries. The dynamical impact of such an intervention has, however, never been formally assessed. Using a combination of mathematical and computational models developed to capture rabies dynamics in the context of control interventions guided by contact tracing of

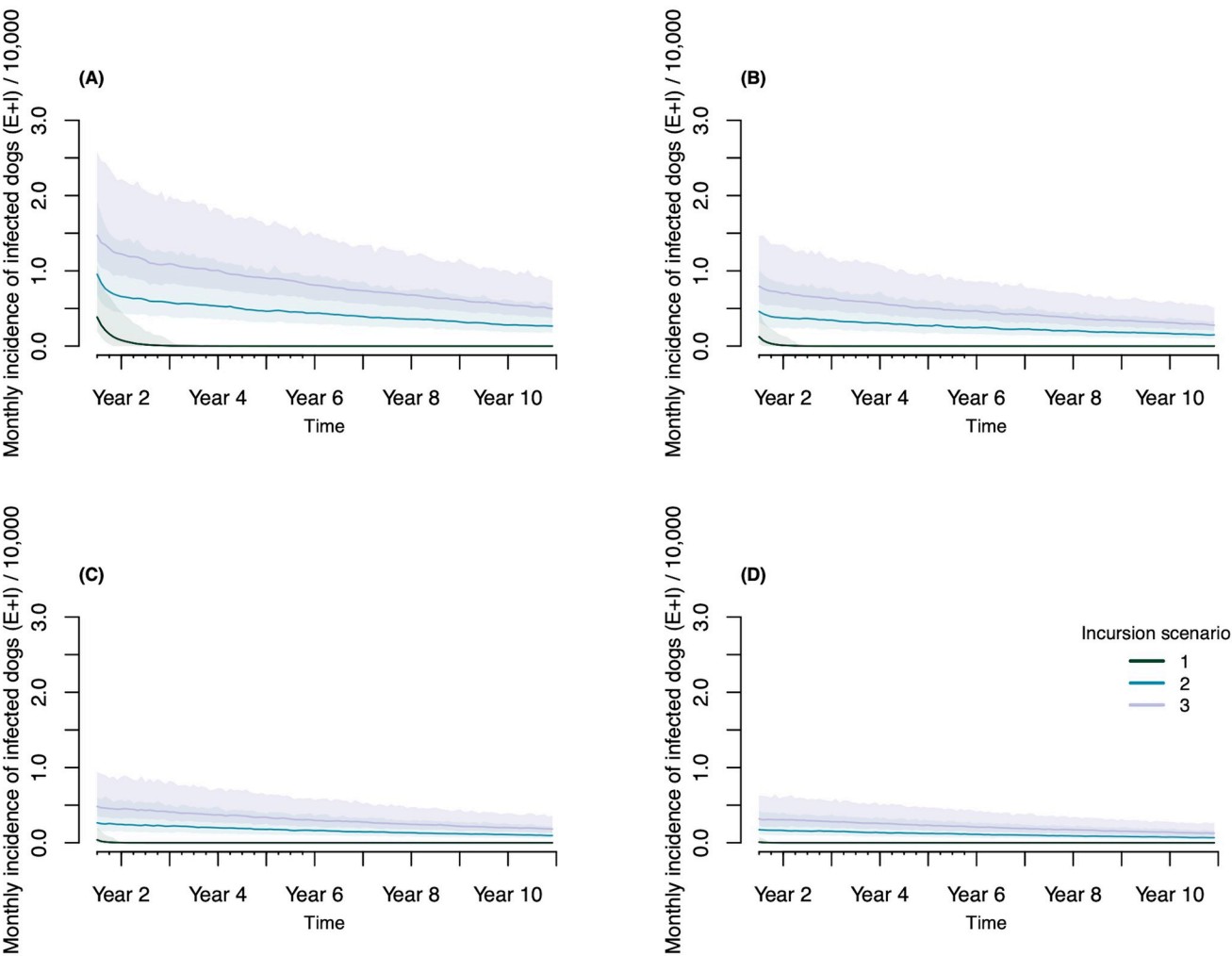

**Fig 7. Example time series for $R_0$ = 1.3 demonstrating rabies dynamics across increasing vaccination coverage levels and incursion probabilities.**
Incursion scenarios 1, 2, and 3 stand for $\bar{i} = 0$, $\bar{i} = 1.5$ and $\bar{i} = 3$ respectively. Vaccination coverage of 0%, 25% 50% and 75% shown in (A), (B), (C) and (D) in the same order. Regardless of the vaccination rate, complete elimination of the virus from the population occurred in the absence of incursions. However, to achieve elimination with incursion probabilities >1.5, high levels of vaccination coverage and extended time lines are to be expected. Shading around projected trajectories indicate the 95% confidence envelope. Note that the time series omit the model burn-in period of the initial six months.

dogs informed by patient bite-histories, here we investigate the impact of quarantine and vaccination on the stability of the system with potential application to other low incidence diseases.

We found that in the deterministic settings even medium levels of quarantine of infected dogs pose a strong pressure on the stability of the system, and in combination with minimal vaccination efforts quarantine would lead to a complete elimination of the disease. Analytical models are powerful tools to explore global dynamics of a system and its long-term evolution, but the insights are relevant only when assessed qualitatively. As such, the theoretical findings suggest that introducing formal quarantine into the rabies management strategies alongside mass dog vaccination campaigns would result in a reduction in the overall burden of rabies cases whilst potentially providing a critical stopgap in areas where immunity coverage falls temporarily below the optimal levels. However, the exact parameter thresholds for when elimination can be expected are only conceptual and will be modulated in empirical settings.

In fact, in the expanded probabilistic SEIRVQ framework the temporal exclusion of dogs through quarantine does not result in elimination of the pathogen in spite of higher vaccination coverage levels than suggested by the analytical model. Stochastic extinctions of individual transmission chains offset by re-introductions of rabies from outside the population via exogenous incursions create a highly non-linear landscape of transmission, requiring more extensive efforts than predicted deterministically given the probabilistic nature of individual transmission events (e.g. barriers to achieving elimination of polio in [38]). Demography may also play an important role; fast population turnover due to high vital rates leads to constant restructuring of the dog population and its immunity profile. Particularly in areas where the dog population undergoes a substantial demographic change, annual high coverage achieving mass dog vaccination campaigns are essential to countervail the immunity loss due to removal of vaccinated dogs and their replacement with susceptible puppies [5, 6].

It is, however, important to note that the depletion of the susceptible population is not associated with rabies, suggesting that changes in the size of the dog population alone will not affect rabies transmission unless accompanied by additional measures in a holistic manner. Given the nature of rabies transmission dynamics, previous research has typically utilised frequency dependence for modelling purposes [6, 32, 45, 46]. Both empirical data and models demonstrating that canine rabies circulates at low prevalence around the world regardless of the size of the dog population [32] indicate that rabies transmission between dogs occurs mostly independently of the population density under a vast range of conditions, implying that a rabid dog will produce on average the same number of infected contacts in a population of any realistic size [2, 6, 9].

Conversely, contacts leading to disease transmission are largely context specific, and they will change as the interventions are being implemented and in response to the phase of the epidemic curve. Social, cultural, environmental and incidental backgrounds can vary widely even across small spatial ranges, resulting in many loosely connected metapopulations that act, for the most time, as individual foci of infection [47, 48]. For diseases with higher transmission rates, smaller scale differences can be averaged across larger spatial aggregates/population, whilst for the lower incidence infections detailed spatial models provide partial leverage in capturing some of the system's heterogeneities.

For example, a variability in the incubation and infectious period distributions can dramatically alter the characteristics of rabies outbreaks depending on the immunity profile of a given setting. Populations in which susceptibility levels fall low (and in an erratic manner) as a result of inconsistent vaccination efforts are more sensitive to such endogenous incursions, resulting in higher case burdens and longer transmissions chains [47]. Similarly, when spatially structured and coupled with heterogeneous immunity backgrounds, exogenous incursions may pose even stronger pressure than observed in our model, leading to further elimination delays and increased control requirements. Whilst critical to answer such questions, extensive spatial models can be costly and intractable in terms of deriving generalizable results across settings and require robust parameterization efforts. In addition, investigations into the biological drivers of variation in disease transmission beyond population-level factors are likely to provide promising insights; however, these are yet to be captured formally in a mathematical framework.

Our findings add onto the existing body of information on rabies management including actionable guidelines and tools supported by decades of operational research, and offer a deeper understanding of the principles and effectiveness of quarantine on rabies dynamics. Implementing contact tracing and quarantine of suspect and probable dogs may bring enormous benefits to public health and the affected communities, particularly in lower vaccination settings challenged by exogenous incursions, directly reducing the number of canine and

human deaths due to rabid bites. Nonetheless, while active investigations and quarantine appear a powerful component of the One Health response in curtailing transmission, large-scale vaccination of dogs is necessary for complete interruption of transmission of the virus and sustained elimination of rabies, given the enduring risk of re-introductions from neighbouring populations [49–51]. With the aspiration to eliminate dog-mediated human rabies by 2030, we conclude that a successful outcome depends on a combination of complementary intersectoral control measures, integrating and building upon operational capacities of both public health and veterinary sectors.

## Acknowledgments

The authors are especially grateful to Ed Hill for useful feedback and comments on the manuscript and figures, as well as to Brad, SB, Clint, Clay, AJ and the rest of the Cordova group for providing a solid support network when needed.

## Author Contributions

**Conceptualization:** Kristyna Rysava, Michael J. Tildesley.

**Formal analysis:** Kristyna Rysava.

**Funding acquisition:** Michael J. Tildesley.

**Investigation:** Kristyna Rysava.

**Methodology:** Kristyna Rysava, Michael J. Tildesley.

**Supervision:** Michael J. Tildesley.

**Visualization:** Kristyna Rysava.

**Writing – original draft:** Kristyna Rysava.

**Writing – review & editing:** Kristyna Rysava, Michael J. Tildesley.

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
