## [Decision Letter · Decision Letter 0]

17 Jul 2023

Dear Dr Rysava,

Thank you very much for submitting your manuscript "Identification of dynamical changes of rabies transmission under quarantine: community-based measures towards rabies elimination" for consideration at PLOS Computational Biology.

As with all papers reviewed by the journal, your manuscript was reviewed by members of the editorial board and by several independent reviewers. In light of the reviews (below this email), we would like to invite the resubmission of a significantly-revised version that takes into account the reviewers' comments.

Please pay special attention to the major concerns raised by Reviewers 1 and 2.

We cannot make any decision about publication until we have seen the revised manuscript and your response to the reviewers' comments. Your revised manuscript is also likely to be sent to reviewers for further evaluation.

Sincerely,

Alex Perkins

Academic Editor

PLOS Computational Biology

James O'Dwyer

Section Editor

PLOS Computational Biology

Reviewer's Responses to Questions

**Comments to the Authors:**

Reviewer #1: In this study, Rysava and Tildesley develop a deterministic and stochastic modeling framework to study the effect of vaccination and quarantine in the context of rabies, a low prevalence endemic disease. The text is well written and the topic is of obvious importance. However, I feel that the analysis falls short in several respects, and would suggest that it be expanded upon prior to publication. My specific points are as follows:

Larger points

-Particularly your stochastic model provides a powerful tool for looking at probabilistic events, and yet it doesn’t feel that this is taken advantage of. You refer repeatedly to the idea of population incursions from other populations vs “endogenous” ones arising from the long incubation period that may outlast the quarantine period, but you don’t explicitly characterize this. I would reformulate Figures 5, 6, and Table 2 in order to see how you can really take advantage of your stochastic model. Can you vary the population size (dogs and humans)? For instance, right now when you report “number of infections” as in Figure 4, it’s not a very meaningful value since the size of the population is not contextualized. Maybe you can show how much disease introduction needs to occur from outside the population as a function of population size, given the possibility of these “endogenous” infections from exposed but not yet infectious individuals?

-I think details of the stochastic model are missing. Can you explain more about how you implement the contact tracing for bites and the subsequent decision to quarantine and observe for symptoms? I know you describe the “overall” quarantine approach by scenario but I think more information is needed.

-I’m having a hard time understanding Table 2. Are these fits or just scenarios you’ve constructed with chosen values? First of all, I thought scenario 3 was a stricter quarantine scenario than scenario 2, yet all values for quarantine rates are lower? Why do the vaccination rates in dogs and humans systematically decrease with increasing R0? And what does “Quarantine incident rate (humans)” mean? Is that the incidence of infection in humans? I didn’t think humans underwent quarantine in this model. You say that Table 2 lets you see the effect of quarantine and vaccination rate, like Figure 5, but I don’t find that obvious at all. All the rates seem to co-vary, and I also don’t know what “incident rate” for quarantine or vaccination means.

Smaller points / questions

-I think it would help to justify the use of many different distributions for parameters, i.e. truncated normal for weekly proportion of bite-injury patients, quarantined dogs (Conway-maxwell-poisson), etc…

-Your figures tend to be very hard to read (axis text and labels too small). In Figure 4, you could also add a label to the color bar for ease of understanding. Figure 5 has the same problems with readability as Figure 4. In some of the smaller plots, the y axis scale makes it such that it’s very hard to get anything quantitative out of the values for quarantine scenarios 2 and 3. In general I think some time could be spent improving their appearance.

-In Figure 4, providing “number of infected dogs” feels a bit out of context, since we don’t know anything about the population size we are working with. For instance you say >2100 infected dogs is more than empirical evidence, but for what size of population? Also the scale bar looks like the value in the bottom left corner of Figure 4B is closer to 3000?

Reviewer #2: The authors describe a new method to model potential interventions to reduce rabies transmission in communities. The paper is largely well-written and the topic is of interest to the rabies modeling community. However, there are several critical issues with the methods and interpretations that would likely take significant effort to address (see below for more detail). Furthermore, the paper is not very helpful to rabies program managers without consideration to the feasibility of implementing quarantine programs to the levels that this analysis claims would be impactful. An accompanying analysis (even simple) of the cost and human resources necessary to operate a program that find 45% of exposed dogs (impossible, in my opinion) is necessary to bring this manuscript out of the realm of theory and actually show prospect for field-level impact.

MAJOR CONCERNS:

1) the authors are incorrectly defining "quarantine" in this model. There are no recommendations for a 14-day quarantine period. There is a 10-day post-bite observation period that is recommended by WHO and the purpose of this is only intended to assess if the biting dog (or cat) was infectious at the time it exposed a human. Dogs that are unvaccinated and exposed to another rabid dog are recommended to undergo a 4-month quarantine period, which is reflective of the rather long and variable incubation period in dogs. More details on this concern can be found below.

2) the authors cite 3 papers to make the bold statement that dog-mediated rabies is NOT density dependent. It should be noted that some of the most commonly cited and respected models in the rabies community have inherent density-dependent assumptions, including Coleman and Dye, which gives us the oft-cited "70%" recommendation and preceding transmission models by Zinsstag and Borse. The idea that rabies is NOT density dependent stems from several heavily biased and misinformed studies and is in direct conflict with all empirical evidence of rabies reductions after implementation of dog population management programs (e.g. Human Society programs to remove unwanted free-roaming dogs from communities). Furthermore, there are numerous wildlife rabies studies demonstrating population density dependence for transmission; it is difficult to believe that such a conserved viral genome would have no density dependence in free roaming dogs. The misunderstanding of density dependence and rabies stems from poor definitions of the susceptible dog population (clearly defined by Coleman and Dye) - when we only consider the dogs that are able to roam freely (at least part of the time) and become expose to rabid dogs, then there is strong evidence for density dependent transmission, as we see in wildlife systems as well.

3) As mentioned above, the lack of consideration to the logistical needs to implement the high level of quarantine identified in this model makes any real-world implications difficult to see happen.

4) The authors should be more clear about their assumption as to how these exposed dogs will be identified. The vast majority of rabid dogs will not bite people, therefore would not be identified in a bite-based triage system as described in this manuscript. So there are 2 populations of dogs that need to be considered: 1) dogs that get rabies and do NOT bite people and 2) dogs that get rabies and bite people (or are part of that traceback program). When the authors refer to 45% quarantine rates, which dog population are they referring to?

SPECIFIC COMMENTS:

INTRO

Line 10: Incomplete sentence ending in “… by;”

METHODS

There is a critical flaw in the logic of this model. The recommended quarantine time for an EXPOSED dog is 4 – 6 months, to ensure that the dog does not develop rabies. The 14-day quarantine (which is actually 10 days, per WHO recommendations) is only to determine if a BITING dog is currently infectious with rabies virus, and informs PEP decisions for the exposed PEOPLE. If the goal of this model is to identify free roaming dogs that are exposed to rabies virus, then I highly doubt the conclusions will be relevant, as these exposures typically go unrecognized by dog owners and community members. However, per the methods described, if the intention is to assume these exposed dogs CAN be identified, then the appropriate parameter value should be 4 to 6 months. https://www.cdc.gov/rabies/specific_groups/veterinarians/potential_exposure.html#:~:text=If%20the%20owner%20is%20unwilling,or%206%20(ferrets)%20months.

Line 114: This sentence could be misleading when describing “n”. Finding dogs exposed after a HUMAN exposure is identified is fairly straightforward. However, many rabid dogs never bite a person and many bite victims never seek medical care necessary to initiate the triage process. Therefore, “n” as described here is only representative of a small proportion of the dogs that drive enzootic transmission.

Line 114: Please provide citations for this “field observations” comment. If this refers to the studies from pastoral Tanzania, numerous other literature has been published in the last decade that conflicts with the observations from this study area, and likely the observations here are non-generalizable.

Line 114: I don’t see where the model accounts for the proportion of exposed dogs that are not involved in a human exposure (from field observation, this is the vast majority of dogs). Where does the model account for this major driving of transmission, which would never be visible to the triage system that this model relies upon for the quarantine process?

Line 115: Critical flaw – 14-days is not the correct parameter value. Exposed dogs, which are unvaccinated, are recommended to be quarantined for 4 – 6 months, which is estimated to reflect >95% of the incubation period. 14-days (which is actually 10) is for biting dogs and is only to ensure they were not shedding virus at the time a person was exposed. This paper has some parameter values that the authors could consider: Negligible risk of rabies importation in dogs thirty days after demonstration of adequate serum antibody titer - PMC (nih.gov) – While the author’s rationale for 14 days is incorrect, this paper does provide support that the immediate vaccination and quarantine of exposed dogs would only require a 30-day quarantine period (rather than 4 months). While official recommendations are still 4 months, scientific evidence for 30-day post-exposure quarantine periods is growing.

Line 116: please use consistent language when referring to EXPOSED vs CASE patients. I think in this sentence the authors mean “exposed”, rather than rabies cases.

RESULTS:

Line 120: Ro is almost certainly related to the density of SUSCEPTIBLE dogs in the community. Rather than randomly allowing the model to reflect a wide range of Ro, why not assume a level of density-dependence to reflect affected communities?

Line 122: Please be more descriptive in talking about the quarantine rates. Are these the rates of ALL EXPOSED DOGS that would be quarantined (45% is unimaginably high), or is this the fraction of dogs that are identified through contact tracing that are found and placed into quarantine?

Line 123: the authors should be aware of several papers (Japan, notably) documenting Ro much higher than 2. However, I think a consideration for past and this current model is that Ro is not a stable value for rabies, and Rt is a more appropriate value to consider. Rt can vary drastically based on the stage of the epidemic and the density of susceptible dogs.

Paragraph 134: The manuscript is not very clear when it comes to explaining if the model is only accounting for quarantine of the proportion of dogs that are involved in a human bite incident, or if they consider all dogs that participate in enzootic transmission. A large proportion of dogs will develop rabies and never bite a person – they will drive transmission regardless of how robust the human bite triage system operates. What assumptions (and these would be critical) did the authors make in regards to the proportion of rabid dogs that actually bite people (human-dog transmission rate)?

A model that has been published several times accounts for this parameter and estimates this dog-to-human transmission rate at 0.00003 km3/dogs/week, and later updated to 2.34*10^-5 by Kunkel et. al. These values are not directly transferrable to the modeling method the authors have chosen, but they highlight the longstanding recognition that the majority of rabid dogs do not expose people. If human bites are the impetus for quarantine in this model, then it needs to better explain how it accounts for the proportion of dogs that do not expose people but drive enzootic transmission in a community.

https://www.ncbi.nlm.nih.gov/pmc/articles/PMC5988334/#pntd.0006490.s001

The urgency of resuming disrupted dog rabies vaccination campaigns: a modeling and cost-effectiveness analysis - PMC (nih.gov)

Paragraph 135: these findings are difficult to believe, given that the quarantine period selected for the model is incorrect. Furthermore, as mentioned, it needs to be more clear what total proportion of exposed dogs the authors are calculating.

Paragraph 138: this is just completely false. The 70% value determined by Coleman and Dye assumes density dependence. The three papers cited for this statement are well-known to be poorly collected data and at least one incorrectly defined susceptible dog populations. Perhaps the most well-cited rabies transmission model (Rabies econ) assumes density dependence, and defines SUSCEPTIBLE dogs as those that are able to roam freely at least part of the time. When models do not consider the confinement status of dogs, then density dependence associations become weak; this is a limitation of the assumptions in those models. The Morters paper has not stood up well over time, as there are many empirical examples where the increased confinement and reduction of dog populations through human measures has drastically reduced rabies cases.

Overall: the authors should consider including an element of feasibility into the analysis; particularly feasibility of the health system resources to identify and quarantine 45% of ALL rabies exposed dogs… (impossible) and the cost to operate such a system. Would these all be in-home quarantines? If so, do we anticipate 100% owner compliance? Would these be at specialized facilities? If so, the cost alone would make this approach impossible for nearly all LMICs.

Reviewer #3: I found this an interesting and mostly clear read, addressing the effectiveness of quarantine for rabies and rabies like diseases. I think this modelling work is useful and interesting. I have only relatively minor comments.

Points to consider (line numbers are not working quite right so referening location as best I can)

L3: what is meant by ‘deceivingly easy’, that contact tracing is harder than it seems? Or not as effective as might be expected in leading to parameter estimates? Or something else?

L7: ‘the system’ is a bit vague, what exactly is being referred to by this term?

L32: ‘Both’? It isn’t clear what the two ‘things’ are?

L35: ‘particularly effective for highly transmissible diseases’ – is this a quantifiable fact? It seems like this might be hard to establish.

L46: I’d think the reason quarantine is rarely applied for human to human transmitted diseases is that there are all sorts of issues with its enforcement and societal compliance, rather than a lack of evidence its effective.

L51: perhaps ‘, we are specificially interested .. ‘

L75: The delta terms would benefit from a bit more explanation. Perhaps comment on the assumptions underlying this formulation of density dependence?

L95: Perhaps indicate this is the Jacobian evaluated at the endemic equilibrium

L96: Does the assumption that N is fixed equate to b = d? Indeed .. is N fixed as a result of the delta term even if b > d? How restrictive is the assumption that N is fixed?

I found the description of the computational model unclear/confusing in a number of places. As I understand it, the human model is included only for the purposes of contact tracing, and the model considers PEP administration only as a by-product of attending a clinic and therefore becoming potentially involved in contact tracing? It feels a bit odd – given the way the model is described - that there is no discussion at all of the impacts of quarantine on human health outcomes although I recognize this could be regarded as out of scope. But if it is out of scope, perhaps focus more on the human dimension as necessary for the purposes of contact tracing, and not health seeking behaviour.

There are rather a lot of ‘In fact’ and ‘As such’ phrases that seem to serve no purpose.

Between L105 and L106: I’m not sure that it’s true that deterministic models cannot be applied to open population dynamics. Of course, it would require that flows ‘in’ and ‘out’ were modelled deterministically.

Between L108 and L109: I’m confused by the negative binomial model here. Different k’s for humans and dogs? I’m not sure how this works. One NB distribution for dog bites and another for dog human bites? In which case what are the respective R0 values? Perhaps some more clarity here.

Between L110 and L111 : ‘The true rate at which a location receives an incursion’ … the use of location here risks confusion as this is not a spatial model as I understand it.

Between L112 and L113: the introduction of ‘case’ and ‘non-case’ incidents would benefit from more explanation.

Between L120 and L121: Presumably the destabilization of the endemic equilibrium point coincides with it ceasing to exist in the positive quadrant? I wonder if from an epidemiological point of view, focusing on I* might be more informative than its stability since the ‘return time’ to the endemic equilibrium is seemingly not of interest.

Between L122 and L122: ‘However, to maintain disease endemicity for R0 = 1.3 in the absence of both interventions, the required number of Infected dogs (I∗) in the population reaches significantly higher values than what is suggested by empirical evidence (> 2110).’ I didn’t understand this remark. I think you are saying that when R0 = 1.3 and there are no control measures in place, the model predicts unrealistically high prevalences? Perhaps rephrase avoiding the use of ‘required’ and stating what the prevalence is since 2110 doesn’t convey this point so clearly. What is the carrying capacity (K) .. perhaps state this in the main text.

Between L127 and L128: I’m curious as to exactly why the stochastic model makes different predictions about elimination. Is this because there is an incursion process in the stochastic model but not the deterministic model? (what happens if the incursion process is ‘turned off’ in the stochastic model?). Or the stochastic model isn’t run for long enough? Or is it simply the effect of variance around previously fixed means?

Between L132 and L133: It makes sense that contact tracing and quarantine are easier to implement when there are fewer cases. But in your study you use quarantine to address an endemic situation – albeit a low prevalence pathogen. Perhaps add this ‘third’ condition (endemic pathogens persisting at low prevalence)?

Fig 5 is painfully hard to see. Legend ‘range of R0 values’. Perhaps dispense with the ‘please’.

Fig 6. I was puzzled by the seeming regularity in the amplitude of the oscillations. Is this a function of strict density dependence?

Add a statement indicating where the code can be found.

**Have the authors made all data and (if applicable) computational code underlying the findings in their manuscript fully available?**

Reviewer #1: Yes

Reviewer #2: Yes

Reviewer #3: Yes

PLOS authors have the option to publish the peer review history of their article (what does this mean?). If published, this will include your full peer review and any attached files.

Reviewer #1: No

Reviewer #2: No

Reviewer #3: No
---

## [Decision Letter · Decision Letter 1]

10 Oct 2023

Dear Dr Rysava,

Thank you very much for submitting your manuscript "Identification of dynamical changes of rabies transmission under quarantine: community-based measures towards rabies elimination" for consideration at PLOS Computational Biology. As with all papers reviewed by the journal, your manuscript was reviewed by members of the editorial board and by several independent reviewers. The reviewers appreciated the attention to an important topic. Based on the reviews, we are likely to accept this manuscript for publication, providing that you modify the manuscript according to the review recommendations.

Sincerely,

Alex Perkins

Academic Editor

PLOS Computational Biology

James O'Dwyer

Section Editor

PLOS Computational Biology

Reviewer's Responses to Questions

**Comments to the Authors:**

Reviewer #1: Overall much improved and addresses my comments. I have a few minor questions remaining that it would be nice to address, but should not impact the decision for publication:

-Text is much better in new Figure 4. The caption makes it seem like B and C should be comparable / same parameters, yet the R0 value used is different in the left panels of both – is this intentional? Also, it might be worth pointing out the different colormap scale. Right now if you looks quickly it seems like exposed incidence is lower than infection, but this is because the R0 is lower, and also it’s a different color scale.

-Why do all the time series seem to just uniformly decrease? There doesn’t seem to be an “endemic” cycle (Figure 7). It seems like if you waited longer than 10 years, incidence may die out in all scenarios. I guess there are no seasonal drivers in the model, but I’m still a bit confused by this. Related, in Figure 6 when you plot monthly incidence by scenario, are you just plotting every single cumulative monthly value? So in reality is a range of numbers that decreases in time that you are plotting?

-In Table 2 caption, what is “unit increase in vaccination effort”? Does that mean a 1% increase in the vaccination rate?

Reviewer #3: I'm satisified that my original points have been addressed.

My only additional suggestion is that the authors provide a reference to the tau leap methodology they refer to.

**Have the authors made all data and (if applicable) computational code underlying the findings in their manuscript fully available?**

Reviewer #1: Yes

Reviewer #3: Yes

PLOS authors have the option to publish the peer review history of their article (what does this mean?). If published, this will include your full peer review and any attached files.

Reviewer #1: No

Reviewer #3: No

Figure Files:

Data Requirements:

Reproducibility:

References:

---

## [Editor Report · Decision Letter 2]

13 Nov 2023

Dear Dr Rysava,

We are pleased to inform you that your manuscript 'Identification of dynamical changes of rabies transmission under quarantine: community-based measures towards rabies elimination' has been provisionally accepted for publication in PLOS Computational Biology.

Best regards,

Alex Perkins

Academic Editor

PLOS Computational Biology

James O'Dwyer

Section Editor

PLOS Computational Biology

---

## [Editor Report · Acceptance letter]

11 Dec 2023

PCOMPBIOL-D-23-00779R2 

Identification of dynamical changes of rabies transmission under quarantine: community-based measures towards rabies elimination

Dear Dr Rysava,

I am pleased to inform you that your manuscript has been formally accepted for publication in PLOS Computational Biology. Your manuscript is now with our production department and you will be notified of the publication date in due course.

With kind regards,

Zsofi Zombor
